# *miR-29b* Regulates TGF-β1-Induced Epithelial–Mesenchymal Transition by Inhibiting Heat Shock Protein 47 Expression in Airway Epithelial Cells

**DOI:** 10.3390/ijms222111535

**Published:** 2021-10-26

**Authors:** Jae-Min Shin, Joo-Hoo Park, Hyun-Woo Yang, Jee Won Moon, Heung-Man Lee, Il-Ho Park

**Affiliations:** 1Department of Otorhinolaryngology-Head and Neck Surgery, Korea University College of Medicine, Seoul 08308, Republic of Korea; shinjm0601@hanmail.net (J.-M.S.); aka_yc@naver.com (J.W.M.); lhman@korea.ac.kr (H.-M.L.); 2Upper Airway Chronic Inflammatory Diseases Laboratory, Korea University College of Medicine, Seoul 08308, Republic of Korea; pjh52763@korea.ac.kr (J.-H.P.); yhw444@korea.ac.kr (H.-W.Y.); 3Medical Device Usability Test Center, Korea University Guro Hospital, Seoul 08308, Republic of Korea

**Keywords:** microRNA, heat shock protein 47, epithelial–mesenchymal transition, transforming growth factor beta-1, tissue remodeling, primary nasal epithelial cells

## Abstract

Tissue remodeling contributes to ongoing inflammation and refractoriness of chronic rhinosinusitis (CRS). During this process, epithelial-mesenchymal transition (EMT) plays an important role in dysregulated remodeling and both *microRNA (miR)-29b* and heat shock protein 47 (HSP47) may be engaged in the pathophysiology of CRS. This study aimed to determine the role of *miR-29b* and HSP47 in modulating transforming growth factor (TGF)-β1-induced EMT and migration in airway epithelial cells. Expression levels of *miR-29b*, HSP47, E-cadherin, α-smooth muscle actin (α-SMA), vimentin and fibronectin were assessed through real-time PCR, Western blotting, and immunofluorescence staining. *Small interfering RNA* (*siRNA*) targeted against *miR-29b* and *HSP47* were transfected to regulate the expression of EMT-related markers. Cell migration was evaluated with wound scratch and transwell migration assay. *miR-29b* mimic significantly inhibited the expression of HSP47 and TGF-β1-induced EMT-related markers in A549 cells. However, the *miR-29b* inhibitor more greatly induced the expression of them. HSP47 knockout suppressed TGF-β1-induced EMT marker levels. Functional studies indicated that TGF-β1-induced EMT was regulated by *miR-29b* and HSP47 in A549 cells. These findings were further verified in primary nasal epithelial cells. *miR-29b* modulated TGF-β1-induced EMT-related markers and migration via HSP47 expression modulation in A549 and primary nasal epithelial cells. These results suggested the importance of *miR-29b* and HSP47 in pathologic tissue remodeling progression in CRS.

## 1. Introduction

Chronic rhinosinusitis (CRS) is defined as the persistent inflammation of the sinonasal mucosa and is a very heterogeneous disorder with unclear pathophysiology that lasts for over 12 weeks [1]. Presently, the majority of CRS patients who fail optimal standard medical treatment are candidates for surgery. However, 10–20% of patients with CRS show only slight improvement in symptoms despite medical and surgical therapy, which is considered refractory CRS [2]. Recent studies suggest that the damage to the physical barrier in the sinonasal epithelium by exogenous agents leads to a dysregulated immune response and pathologic remodeling contributing to disease recalcitrance in CRS [3,4]. Tissue remodeling is a core reaction to stress, such as damage to tissues or chronic inflammation. It is well known that not only in the lower airway diseases, such as asthma or chronic obstructive pulmonary disease, the upper airway diseases, such as allergic rhinitis or CRS are also associated with tissue remodeling [5].

Although the exact mechanism of pathological remodeling in respiratory disease is not fully established, current evidence suggests that it can be related to epithelial-mesenchymal transition (EMT) [6]. EMT is a dynamic process of losing epithelial cell function because of the action of pleiotropic cytokines such as transforming growth factor (TGF)-β1. During the damaged mucosal barrier repair cascades, EMT contributes to pathological remodeling events and leads to refractory CRS [7]. In particular, a recent study demonstrated that EMT-related markers were up-regulated in CRS tissues compared with controls and highly correlated with disease severity in CRS patients [8].

MicroRNAs (miRs) are short non-coding RNAs that perform essential physiological and pathological processes, including wound healing [9]. They regulate target gene expression by destabilizing their mRNA and inducing translational repression. Among them, *miR-29b* is known to modulate wound healing and tissue fibrosis [10]. Recent evidence has shown that *miR-29b* regulates TGF-β1-induced EMT in a pulmonary fibrosis animal model [11].

Heat shock protein 47 (HSP47) is a type of molecular chaperone that contributes to various types of collagen maturation, and it may drive the tissue remodeling and accumulation of extracellular matrix (ECM). In our previous study, we reported the relationship between the increased expression of HSP47 in nasal tissue and CRS severity [12]. Furthermore, Zhu et al. demonstrated that *miR-29b* overexpression inhibits ECM production by modulating HSP47 expression in dermal fibroblasts and potentially contributes to tissue remodeling [13]. Based on these above findings, we hypothesized that *miR-29b* could down-regulate EMT through HSP47, which was related with tissue remodeling in CRS. Thus, the purpose of the present study was to investigate whether *miR-29b* could modulate TGF-β1-induced EMT via HSP47 expression in airway epithelial cells.

## 2. Results

### 2.1. HSP47 Expression, Targeted by miR-29b, Was Induced by TGF-β1 in A549 Cells

To investigate whether *miR-29b* modulates TGF-β1-induced EMT, we analyzed the potential target of *miR-29b* using TargetScan (www.targetscan.org, version 8.0). We accessed this link on 25 October 2021. We found a putative *miR-29b* target site in the HSP47 (SERPINH1) 3′ untranslated region (3′-UTR) (Figure 1a). To evaluate the effect of TGF-β1 on *miR-29b* and HSP47 expression in A549 cells, we measured *miR-29b* and *HSP47* mRNA expression using quantitative real-time PCR (qPCR) in A549 cells treated with TGF-β1 at the indicated doses (0.5, 1, 2.5, 5 or 10 ng/mL, 24 h). TGF-β1 significantly reduced *miR-29b* (Figure 1b) and increased *HSP47* mRNA levels in A549 cells (Figure 1c) in a dose-dependent manner. We also measured the HSP47 protein expression using Western blotting. TGF-β1 significantly induced the expression of the HSP47 protein at 72 h in a dose-dependent manner (Figure 1d). These results indicated that HSP47, a direct target of *miR-29b*, was induced by TGF-β1 in A549 cells.

### 2.2. miR-29b Modulated mRNA and Protein Expression Levels of TGF-β1 in A549 Cells

We determined the effect of *miR-29b* on TGF-β1-induced EMT using qPCR, Western blotting, and immunofluorescence staining. First, the *miR-29b mimic* was transfected into the A549 cells before TGF-β1-treatment. We found that *miR-29b mimic* significantly elevated TGF-β1-reduced *miR-29b* expression (Figure 2a) and inhibited TGF-β1-induced HSP47 expression (Figure 2b). *miR-29b* mimic significantly inhibited the luciferase activity of the wild-type HSP47-3′-UTR. *miR-29b* mimic had no effect on the luciferase activity of the mutant HSP47-3′-UTR (Figure 2c). Transfection of the *miR-29b mimic* resulted in a significant induction of E-cadherin and reduction of HSP47, α-SMA, vimentin, and fibronectin mRNA (Figure 2d) and their protein levels (Figure 2e). Furthermore, we verified these findings through immunofluorescence staining, and the results were similar to those obtained from Western blotting (Figure 2f). Next, the *miR-29b inhibitor* was transfected into the A549 cells. TGF-β1-inhibited *miR-29b* expression was further inhibited by the *miR-29b inhibitor* (Figure 3a), and TGF-β1-induced *HSP47* mRNA expression was further induced by the *miR-29b inhibitor* (Figure 3b). The *miR-29b* inhibitor significantly stimulated the luciferase activity of the wild-type HSP47-3′-UTR, however, the *miR-29b* inhibitor had no effect on the luciferase activity of the mutant HSP47-3′-UTR (Figure 3c). The *miR-29b inhibitor* increased TGF-β1-induced expression of HSP47 and EMT markers mRNA (Figure 3d) and their protein levels (Figure 3e). We also verified these findings using immunofluorescence staining, and the results were similar to those obtained from Western blotting (Figure 3f).

### 2.3. Silencing the HSP47 Inhibited TGF-β1-Induced EMT in A549 Cells

We hypothesized that HSP47 acts downstream of *miR-29b* and verified the effect of HSP47 on *miR-29b* expression by silencing HSP47 using *siHSP47*. The expression of *miR-29b* was inhibited not only by TGF-β1 with *siControl* but also by TGF-β1 with *siHSP47* (Figure 4a). These data implied that *miR-29b* might act upstream of HSP47. TGF-β1-induced *HSP47* mRNA expression was inhibited by *siHSP47* transfection (Figure 4b). To determine whether HSP47 regulates EMT-related markers in A549 cells, we measured the mRNA and protein levels of the EMT markers after *siHSP47* transfection. Transfection of the siHSP47 inhibited TGF-β1-induced HSP47, α-SMA, vimentin, and fibronectin mRNA and protein levels. In addition, TGF-β1-down-regulated E-cadherin up-regulated by the transfection of the *siHSP47* (Figure 4c,d). Additionally, we determined the protein levels of HSP47 and EMT-related markers using immunofluorescence staining, and the results were similar to those of Western blotting.

### 2.4. miR-29b Overexpression and HSP47 Silencing Inhibits TGF-β1-Induced Cell Migration in A549 Cells

Cell migration was examined with wound scratch migration and transwell migration assays. To determine the migration potential, TGF-β1-migrated cells were counted after wound scratch migration and transwell migration assays. The area between the scratch lines was reduced by the TGF-β1, but not by the *miR-29b mimic* or *siHSP47* with TGF-β1 (Figure 5a and Figure 6a). The number of migrated cells at the bottom of the transwell chamber was significantly higher in cells treated with TGF-β1 than in the controls. However, the *miR-29b mimic* or *siHSP47* with TGF-β1 decreased the number of migrated cells compared to the TGF-β1–treated group (Figure 5b and Figure 6b). These results demonstrated that the overexpression of *miR-29b* and silencing of HSP47 result in significant suppression of TGF-β1-induced migration in A549 cells.

### 2.5. Overexpression of miR-29b or Silencing of HSP47 Inhibits TGF-β1-Induced EMT in Primary Nasal Epithelial Cells

In order to check whether the same results as the above experiment are obtainable in primary nasal epithelial cells, we transfected primary nasal epithelial cells with *miR-29b mimic* or *siHSP47*. The *miR-29b mimic* inhibited TGF-β1-induced HSP47 expression. As for EMT related markers, *miR-29b mimic* induced TGF-β1-inhibited E-cadherin expression and inhibited TGF-β1-induced α-SMA, vimentin, and fibronectin expression (Figure 7a). We also determined the expression and localization of E-cadherin and vimentin in primary nasal epithelial cells through immunofluorescence staining (Figure 7b). The *miR-29b mimic* induced TGF-β1-inhibited E-cadherin expression at the cell membrane. However, TGF-β1-induced vimentin expression was inhibited by the *miR-29b mimic* in the cell cytoplasm. The results were similar with the primary nasal epithelial cells transfected with *siHSP47*. (Figure 7c,d).

## 3. Discussion

In this study, we demonstrated that *miR-29b* modulates the protein and mRNA expression levels of EMT-related makers, which are induced by TGF-β1 in airway epithelial cells. Moreover, it has been found to attenuate this process by HSP47 knockout using *si**RNA*. In addition, we showed that TGF-β1-enhanced cell migration was significantly inhibited by the *miR-29b mimic* and *si**HSP47* in A549 cells. To the best of our knowledge, this study provides the first evidence that *miR-29b* suppresses TGF-β1-induced EMT and migration via HSP47 in airway epithelial cells. These results indicate that *miR-29b* and HSP47 are key regulators of TGF-β1-induced EMT in chronic airway inflammatory diseases such as CRS.

To understand the causes of refractory CRS that do not respond to currently available treatment, most investigators categorized CRS by phenotype following nasal polyps [1]. Recently, a paradigm in which CRS has been differentiated into CRS endotypes based on prominent inflammatory cells, such as eosinophils, or specific cytokines, such as IL-4, IL-5, and IL-13 [14]. Moreover, the concept of tissue remodeling has been well established in lower respiratory diseases, such as asthma, that share similar pathological mechanisms with refractory CRS. According to current evidence, tissue remodeling in CRS shows characterized clinical features based on the endotype and continuously occurs during ongoing inflammation [15]. Kao et al. reported further evidence that CRS mucous, regardless of phenotype, demonstrated dysregulations of biological processes related to tissue remodeling using proteomic analysis [16]. Ryu et al. also suggested that EMT and tissue remodeling play key roles in neutrophilic CRS [17]. Taken together, these findings suggest that tissue remodeling may be a common upstream mechanism that leads to downstream manifestations such as endotype and phenotype in CRS. Thus, we focused on pathologic tissue remodeling via EMT, which drives ongoing inflammation and contributes to CRS refractoriness.

Although the precise mechanisms of pathologic tissue remodeling in refractory CRS have not been fully identified, emerging evidence suggests that ECM deposition is also correlated with CRS severity, which is associated with tissue remodeling [18]. HSP47 has been widely accepted as a potent player that is closely related to tissue remodeling, mainly characterized by ECM accumulation, such as fibrosis and collagen deposition in recalcitrant fibrotic disease [19,20]. Our previous study revealed that HSP47 contributed to the tissue remodeling process by ECM production in nasal fibroblasts [12]. Here, we demonstrated that HSP47 mediates the dysregulated remodeling process represented by EMT in airway epithelial cells and that *miR-29b* regulates this pathological process. In addition, EMT is a potential source of fibroblasts, which synthesize the components of ECM [21]. Consequently, HSP47 is receiving significant attention as a multifunctional regulator of both ECM accumulation and EMT during the dysregulated remodeling process in the upper airway, which may be considered as a meaningful therapeutic target for refractory CRS patients.

miRNAs consist of noncoding small RNA molecules of 19–22 nucleotides, which bind to the 3′-untranslated region (UTR) of target genes and regulate the expression of protein-coding genes by degrading target mRNAs or repressing their translation of target genes [22]. A number of miRNAs are reported to be associated with pathophysiology of upper or lower airway diseases [23]. The *miR-29b* family, one of various miRNAs, are involved in the regulation of cancer metastasis, wound healing, inflammation and especially tissue remodeling [24,25]. Previous *miR-29b* studies have reported that *miR-29b* contributes to tissue remodeling-related diseases via EMT in the lower airway [11]. Moreover, Montgomery et al. found that systemic delivery of *miR-29b* mimics inhibited and reversed fibrotic events without side effects in a lung fibrosis animal model [26]. Similarly, Zhu et al. found that local delivery of *miR-29b* lentivirus suppressed tissue remodeling makers via HSP47 in an animal model with scarring, which implies the potential use of *miR-29b* as a local treatment [13]. Consistent with the above findings, our results provide new evidence regarding the inhibitory role of *miR-29b* in the upper airway, especially its role in suppressing pathologic remodeling. By contrast, Li et al. reported that miR-21 effectively suppressed EMT induced by TGF-β1 in human nasal epithelial cells [27]. In other reports, *miR-29b* binds to target the ECM gene, signaling molecules or inflammatory response-related factors [10,25]. As we mentioned above, we focused on HSP47, which mediates multiple functions in the process of dysregulated remodeling. Thus, we preferentially considered *miR-29b* as an experimental target, which is strongly associated with HSP47 expression. We showed that *miR-29b* directly bind to HSP47 target sites and inhibit expression of HSP47. In this study, we consistently revealed that the up- and down-regulation of *miR-29b* expression significantly modulated TGF-β1-induced EMT in airway epithelial cells.

It is well established that the TGF-β1 signaling pathway plays a key role in TGF-β1-induced EMT. The present study identified that TGF-β1 suppressed *miR-29b* expression and that HSP47 expression was up-regulated by TGF-β1 in a dose- and time-dependent manner. These findings are consistent with recent evidence that TGF-β1 down-regulates miR-29 via the Smad3 signaling pathway in a renal fibrosis model [28]. Furthermore, we provide direct evidence that HSP47 expression was significantly modulated by the up- or down-regulation of *miR-29b* expression, establishing the relationship between *miR-29b* and HSP47 in the upper respiratory tract diseases for the first time.

In summary, we found that *miR-29b* down-regulates TGF-β1-induced EMT and cell migration via HSP47 in airway epithelial cells. Therefore, our data may provide valuable information on the potential therapeutic targets of *miR-29b* and HSP47 in patients with refractory CRS. Further basic and clinical studies are crucial to explore more effective management of CRS.

## 4. Materials and Methods

### 4.1. Cell Cultures

The human airway epithelial cell line A549 (ATCC CCL-185) was obtained from the American Type Culture Collection, Rockville, MD, USA. A549 is a cancer cell line, therefore aberrant gene expression and altered molecular pathways could influence the results. Cells were cultured in an RPMI 1640 (Hyclone, Logan, UT, USA) medium supplemented with 10% fetal bovine serum, 1% 10,000 units/mL penicillin, and 1% 10,000 μg/mL streptomycin (Invitrogen, Carlsbad, CA, USA) then incubated at 37 °C and 5% CO_2_. Brushes were used to scrape the mid-part of the inferior turbinate for primary nasal epithelial cell cultures, as previously reported [25], followed by immediate transfer into an RPMI 1640 medium containing 1% penicillin streptomycin. Human primary nasal epithelial cells were cultured in a PneumaCult-Ex Plus Medium (Stemcell Technologies, Vancouver, BC, Canada) for 3 days in type I collagen-coated flasks (Costar, Corning, NY, USA).

Written informed consent was obtained from all the subjects, and this research was conducted in accordance with the Declaration of Helsinki. This study was approved by the Korea University Medical Center Institutional Review Board (KUGH12041-001).

### 4.2. Transfection with Small Interference (si)RNA Targeted against miR-29b and HSP47

The efficacy of transfection reagents in promoting siRNA uptake was investigated. A549 cells and primary nasal epithelial cells were seeded at 6 × 10^5^ cells/well in 6-well culture plates and incubated at 37 °C, during which time the cells adhered to the plastic reaching up to 50–60% confluence. Then the transfection was performed with *miR-29b mimic* (Sigma-Aldrich, St. Louis, MO, USA), *miR-29b inhibitor* (Sigma-Aldrich), or *siHSP47* (1135737, Bioneer, Daejeon, Korea) using LipofectamineTM RNAiMax (Invitrogen) for 24 h according to the manufacturers’ instructions. Then, the cells were stimulated with TGF-β1 for 24 h or 72 h. The *miRNA mimic* (Sigma-Aldrich) was used as the negative control.

### 4.3. qPCR

RNA extraction from the A549 cells was conducted using TRIzol reagent (Invitrogen). cDNA synthesis was performed with 2 µg of total RNA using MMLV and reverse-transcribed (Invitrogen) for a reverse transcription reaction, according to the manufacturer’s protocol. qPCR analysis was performed to evaluate the expression levels of *HSP47, E-cadherin, vimentin, α-SMA*, and *fibronectin; GAPDH* was used as housekeeping controls. Primer sequences are listed in Figure 1a. The expression levels of these mRNAs were analyzed through the QuantStudio 3 Real-Time PCR System (Applied Biosystems, Foster City, CA, USA). The results were normalized to *GAPDH* expression, then shown as a fold ratio over the expression of the control group.

### 4.4. miR-29b Expression Analysis

Total RNA was reverse-transcribed using the miScript Reverse Transcription Kit according to the manufacturer’s instructions with 5x miScript HiFlex Buffer (Qiagen, Valencia, CA, USA), which promotes the conversion of both mature and precursor miRNA. The relative expression levels of mRNA and *miR-29b* were quantified using Power SYBR Green PCR Master Mix (Applied Biosystems) and miScript SYBR Green PCR kit (Qiagen), respectively. After the reaction, the relative mRNA and *miR-29b* levels were calculated based on the Ct values and normalized with *GAPDH* or *U6* in each sample.

### 4.5. Immunofluorescence Staining

The cells were placed on coverslips and exposed to TGF-β1 (5 ng/mL) after transfection with *miR-29b mimic*, *miR-29b inhibitor*, or *siHSP47*. Subsequently, The cells were permeated with 0.2% Triton X-100 in 1% bovine serum albumin for 10 min, blocked by 5% bovine serum albumin for 1 h at room temperature, incubated overnight at 4 °C with monoclonal anti-HSP47, anti-E-cadherin, and anti-vimentin antibodies (Sigma-Aldrich), then fixed with 4% paraformaldehyde. Goat anti-mouse Alexa 488 (Invitrogen) and goat anti-rabbit Alexa 555 (Invitrogen) secondary antibodies were added to the cells for incubation. Finally, the nucleus was counterstained with 4′,6-diamidino-2-phenylindole (Invitrogen) and the stained cells were visualized using a confocal laser scanning microscope (LSM700, Zeiss, Oberkochen, Germany).

### 4.6. Wound Scratch Migration Assay

The A549 cells were seeded into 6-well cell culture plates. The cell culture was allowed to reach approximately 90% confluence, and then placed for 24 h in a serum-free RPMI-1640 medium for serum starvation. Next, the cells were treated with mitomycin C (5 µg/mL; R&D Systems, Minneapolis, MN, USA) for 15 min to prevent cell proliferation. A sterile 200-µL yellow pipette tip (Labcon, Petaluma, CA, USA) was used to make a straight scratch. Cell debris was washed out with the serum-free media and was further transfected with *miR-29b mimic* or *siHSP47* before treatment with TGF-β1 for 72 h. The cells were stained with Diff-Quik (Sysmex, Kobe, Japan), and images of the migrated cells were acquired using an Olympus BX71 microscope (Olympus, Tokyo, Japan). In the micrograph, the area in between the lines refers to the area of the initial wound scratch, and cells identified between the scratch lines were counted with ImageJ analyzer (National Institutes of Health [NIH], Bethesda, MD, USA).

### 4.7. Transwell Migration Assay

Transwell chambers with a pore size of 8 µm were placed in 24-well cell culture plates (Costar). The A549 cells were trypsinized and resuspended in 100 µL of serum-free medium at a concentration of 1 × 10^5^ cells/mL, then seeded in the upper transwell chamber. Next, the cells were transfected with *miR-29b mimic* or *siHSP47*, and 500 µL of RPMI 1640 containing TGF-β1 was added to the lower chamber as a chemoattractant. After incubation for 72 h, non-migratory A549 cells remaining at the upper surface of the chamber were removed with cotton swabs. Thereafter, the cells in the lower surface of the membrane, which migrated through the pores, were stained using Diff-Quik, and images of the visualized cells were acquired. Likewise, 5 fields were randomly selected for imaging with a microscope at ×200 magnification (Olympus BX51; Olympus).

### 4.8. Statistical Analysis

All experiments were repeated at least thrice to provide substantial data. Unpaired t-tests, one-way analysis of variance, and Tukey’s test were conducted to compare the differences between the controls and the experimental conditions. All statistical analyses were conducted using GraphPad Prism software ver. 5 (Graph Pad Software, San Diego, CA, USA), and *p* < 0.05 was considered statistically significant.

## Figures and Tables

**Figure 1 ijms-22-11535-f001:**
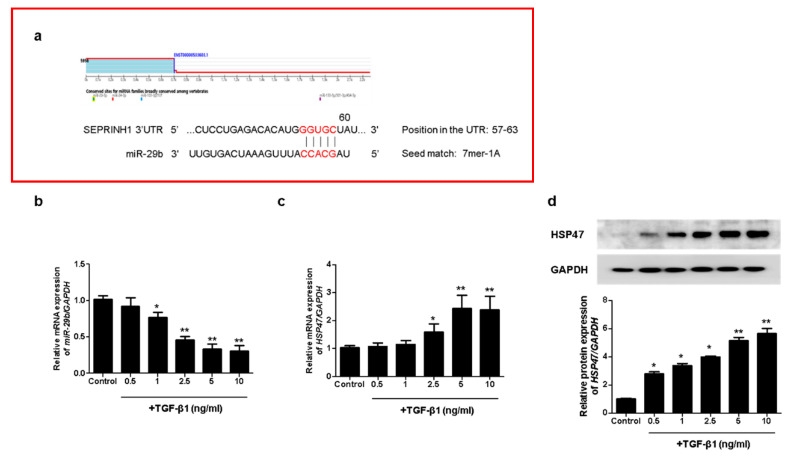
TGF-β1 reduced *miR-29b* and induced HSP47 expression in A549 cells. (**a**) The putative binding sites of *miR-29b* and HSP47 were predicted using TargetScan (www.targetscan.org). A549 cells were treated with TGF-β1 at the indicated doses (0.5, 1, 2.5, 5 or 10 ng/mL, 24 h). (**b**) *miR-29b* and (**c**) *HSP47* mRNA expression were measured using qPCR. A549 cells were treated with TGF-β1 (1 ng/mL) at the indicated dose (0.5, 1, 2.5, 5 or 10 ng/mL, 72 h). (**d**) HSP47 protein expression was measured using Western blotting. Data are expressed as the mean ± SEM of three independent experiments. * *p* < 0.05 vs. control, ** *p* < 0.001 vs. control.

**Figure 2 ijms-22-11535-f002:**
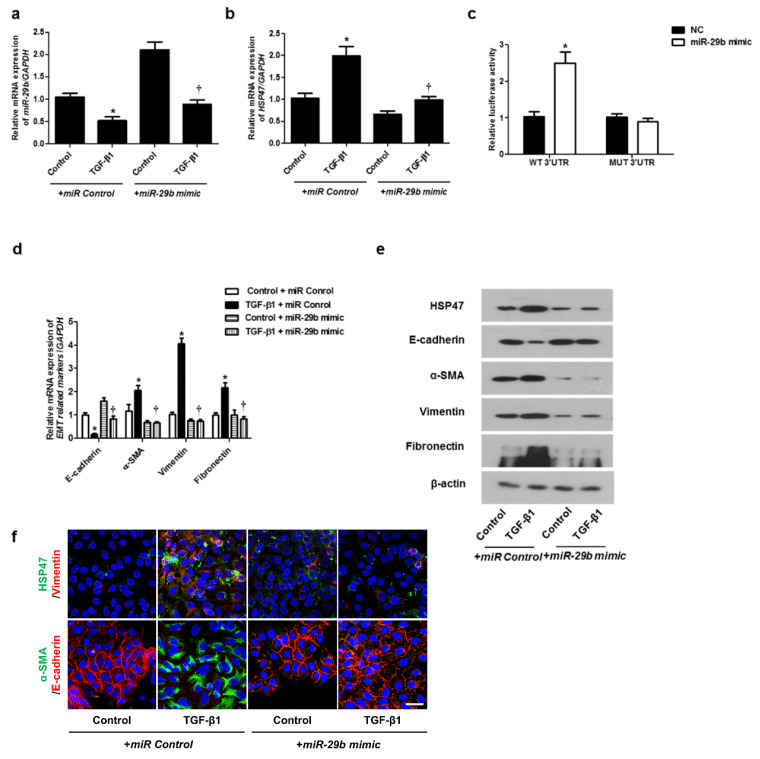
Overexpression of *miR-29b* inhibited mRNA and protein expression levels of TGF-β1-induced EMT markers in A549 cells. A549 cells were stimulated with TGF-β1 (5 ng/mL) with *miR control* or *miR-29b mimic*. (**a**,**b**) mRNA expression levels of *miR-29b* and HSP47 were determined using qPCR. (**c**) HSP47 luciferase activity was measured by luciferase assay. (**d**) *E-cadherin*, *α-SMA*, *vimentin*, and *fibronectin* mRNA levels were analyzed through qPCR. (**e**) Protein expression levels of HSP47, E-cadherin, α-SMA, vimentin, and fibronectin were determined using Western blotting. (**f**) The cells were treated with TGF-β1 for 72 h after transfection of *miR-29b mimic* and then assessed for HSP47 (1st line, green), vimentin (1st line, red), α-SMA (2nd, green), and E-cadherin (2nd, red) expression/localization using immunofluorescence. Nuclei were stained with DAPI (blue). Scale bar = 20 μm. Values are expressed as mean ± SEM of three independent samples. * *p* < 0.05, vs. control + *miR Control*; † *p* < 0.05, vs. TGF-β1 + *miR Control*.

**Figure 3 ijms-22-11535-f003:**
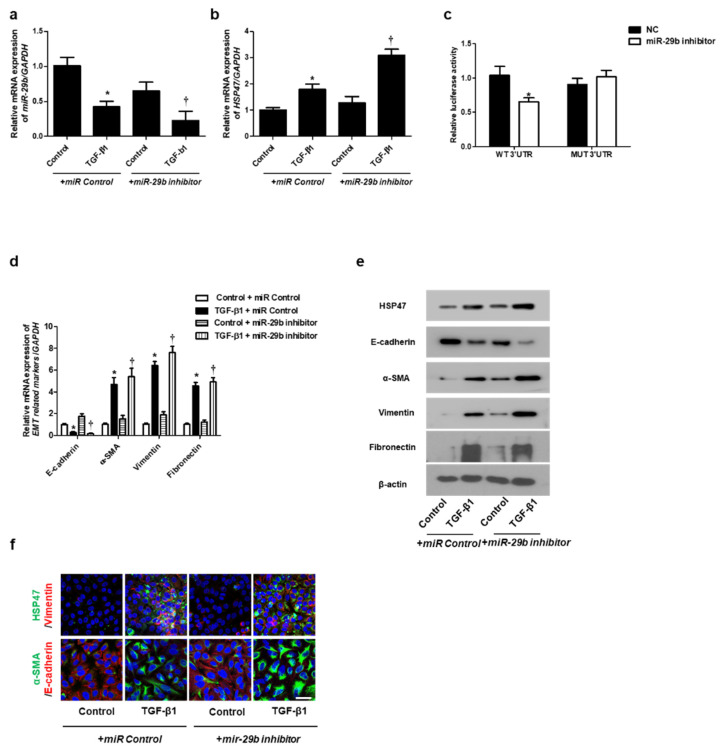
Inhibition of *miR-29b* expression induced mRNA and protein expression levels of TGF-β1-induced EMT markers in A549 cells. A549 cells were stimulated with TGF-β1 (5 ng/mL) with *miR control* or *a miR-29b inhibitor*. (**a**,**b**) The mRNA expression levels of *miR-29b* and *HSP47* were analyzed using qPCR. (**c**) HSP47 luciferase activity was measured by luciferase assay. (**d**) The mRNA levels of *EMT-related markers* were measured using qPCR. (**e**) Protein expression levels of HSP47, E-cadherin, α-SMA, vimentin and fibronectin were determined using Western blotting. (**f**) The cells were treated with TGF-β1 for 72 h after transfection with *miR-29b inhibitor*, and then assessed for HSP47 (1st line, green), vimentin (1st line, red), α-SMA (2nd, green), and E-cadherin (2nd, red) expression/localization using immunofluorescence. Nuclei were stained with DAPI (blue). Scale bar = 20 μm. Values are expressed as mean ± SEM of three independent samples. * *p* < 0.05 vs. control + *miR Control*; † *p* < 0.05, vs. TGF-β1 + *miR Control*.

**Figure 4 ijms-22-11535-f004:**
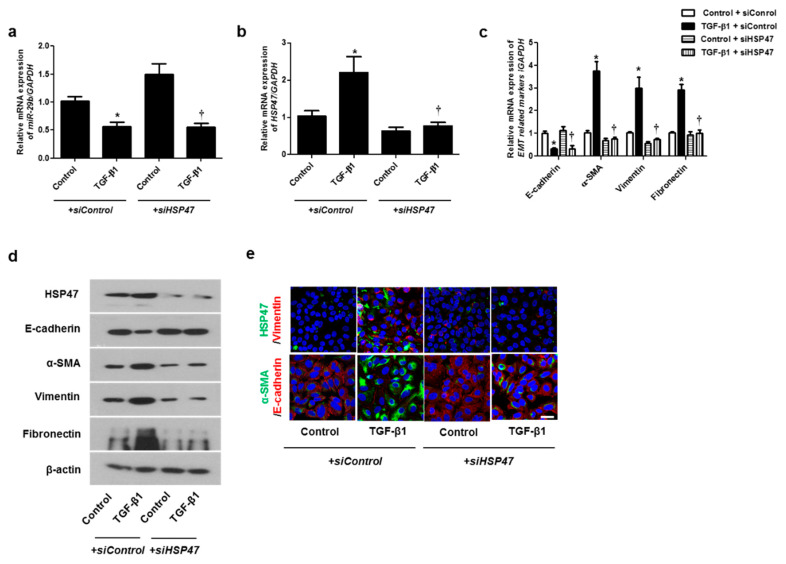
Inhibition of HSP47 induced mRNA and protein expression levels of TGF-β1-induced EMT markers in A549 cells. Specific *HSP47 siRNAs* were transfected prior to treatment with or without TGF-β1 for 24 h in A549 cells. (**a**,**b**) The mRNA expression levels of *miR-29b* and *HSP47* were analyzed using qPCR. (**c**) The mRNA levels of *EMT-related markers* were determined using qPCR. (**d**) Specific *HSP47 siRNAs* were transfected prior to treatment with or without TGF-β1 for 72 h. Protein levels of HSP47 and EMT-related markers were evaluated using Western blotting. (**e**) Confocal laser scanning microscope was used to detect immunofluorescence. Representative fluorescein immunocytochemical staining shows HSP47 (1st line, green), vimentin (1st line, red), α-SMA (2nd line, green), and E-cadherin (2nd line red) with nuclear DAPI (blue). Scale bar = 20 μm. Data are expressed as the mean ± SEM of three independent experiments. * *p* < 0.05, vs. control + *siControl*; † *p* < 0.05, vs. TGF-β1 + *siControl*.

**Figure 5 ijms-22-11535-f005:**
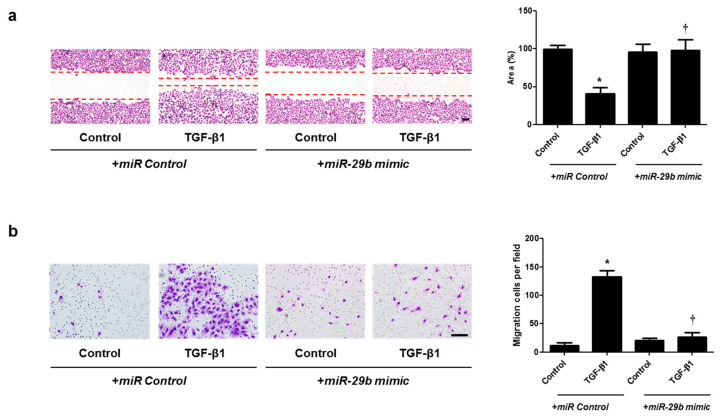
Overexpression of *miR-29b* contributed to A549 cell migration. A549 cells were seeded and transfected with *miR control* or *miR-29b mimic* before treatment with TGF-β1 for 72 h. (**a**) A wound scratch migration assay was performed to investigate the migration of A549 cells. Microscopic observation was performed 72 h after scratching the surface of a confluent layer of cells. (**b**) Inhibitory effects of the *miR-29b mimic* are also shown using a transwell migration assay. Scale bar = 100 μm. Data are expressed as the mean ± SEM of three independent experiments. * *p* < 0.05, vs. control + *miR Control*; † *p* < 0.05, vs. TGF-β1 + *miR Control*.

**Figure 6 ijms-22-11535-f006:**
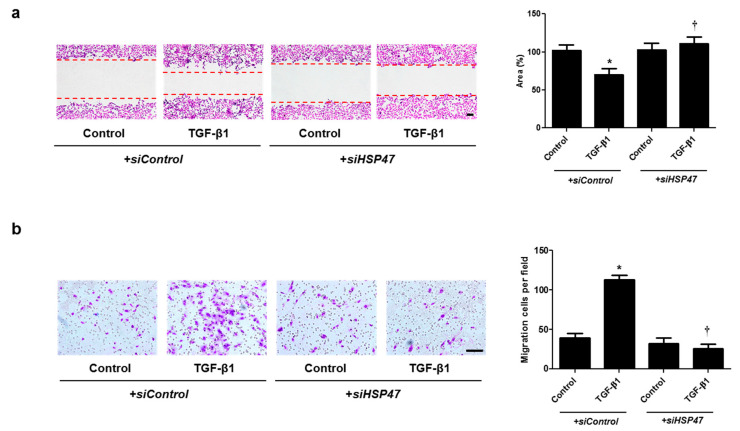
Silencing of HSP47 expression contributed to A549 cell migration. A549 cells were seeded and transfected with *siControl* or *siHSP47* before treatment with TGF-β1 for 72 h. (**a**) A wound scratch migration assay was performed to investigate the migration of A549 cells. Microscopic observation was performed 72 h after scratching the surface of a confluent layer of cells. (**b**) Inhibitory effects of *siHSP47* are shown using a transwell migration assay. Scale bar = 100 μm. Data are expressed as the mean ± SEM of three independent experiments. * *p* < 0.05, vs. control + *siControl*; † *p* < 0.05, vs. TGF-β1 + *siControl*.

**Figure 7 ijms-22-11535-f007:**
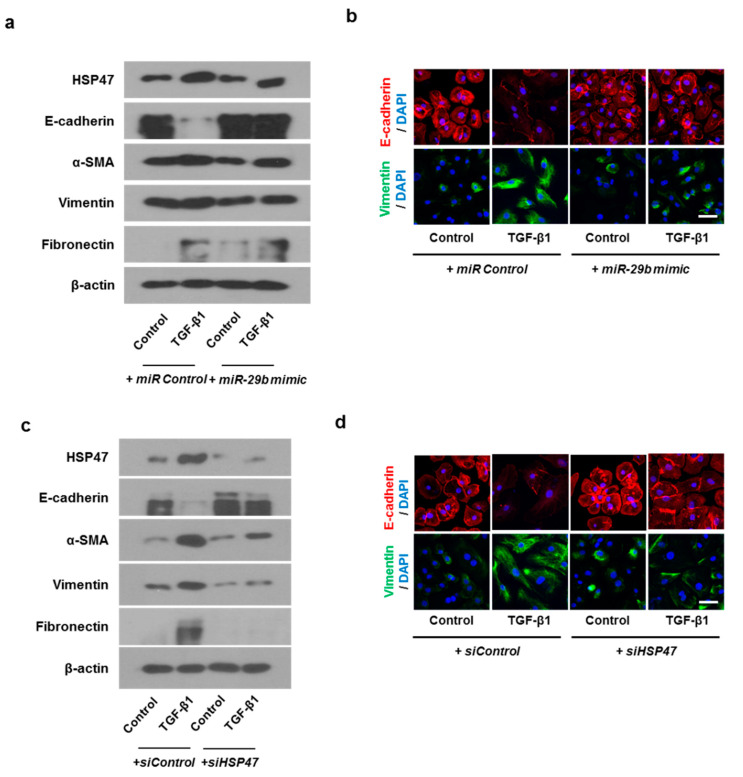
Overexpression of *miR-29b* and silencing of HSP47 inhibited EMT induced by TGF-β1 in primary nasal epithelial cells. Primary nasal epithelial cells were cultured and pretreated with or without *miR-29b mimic* and *siHSP47* before stimulation with TGF-β1 for 72 h. (**a**,**b**) Effect of *miR-29b mimic* on HSP47, E-cadherin, α-SMA, vimentin, and fibronectin protein expression in TGF-β1-stimulated primary nasal epithelial cells, as determined by Western blotting and immunofluorescence. (**c**,**d**) Effect of *siHSP47* on HSP47, E-cadherin, α-SMA, vimentin, and fibronectin protein expression in TGF-β1-stimulated primary nasal epithelial cells determined by western blotting and immunofluorescence. Representative fluorescein immunocytochemical staining shows E-cadherin (red) and vimentin (green) with nuclear DAPI (blue). Scale bar = 20 μm. Data are expressed as the mean ± SEM of three independent experiments.

## Data Availability

The data presented in this study are available on request from the corresponding author.

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
