# Peer review of "miR-29b Regulates TGF-β1-Induced Epithelial–Mesenchymal Transition by Inhibiting Heat Shock Protein 47 Expression in Airway Epithelial Cells"

_ijms, 2021, doi:10.3390/ijms222111535_

Round 1

Reviewer 1 Report

The authors examined the role of miR-29b and HSP47 in regulating EMT induced by TGF-B1 in epithelial cells. The motivation of the study is to understand how EMT is regulated in the context of chronic rhinosinusitis (CRS). The authors have indicated that TGF-B1 reduced the level of miR-29b and induced level of HSP47 in a dose dependent manner. However, the authors did not show direct regulation of miR-29b of HSP47. The authors then did gain-of-function and loss-of-function of miR-29b and showed that miR-29 perturbation affects the mRNA and protein expression levels of TGF-B1 induced EMT markers in A549 cells. Similarly, they showed that loss of function of HSP47 (proposed direct target of miR-29b) had similar effect as miR-29b gain-of-function on TGF-B1 induced EMT markers. Although built in these experiments, results of experiments that only show TGF-B1’s effect on EMT markers would make the logical flow of the manuscript better (ie. as part of Fig. 1). Overall, they showed that TGF-B1 is upstream of miR-29b and HSP, which are upstream of EMT markers such as vimentin, a-SMA, and E-cadherin. Importantly, increasing miR-29b or decreasing HSP levels inhibited migration of A549 cells as well as in primary nasal epithelial cells.

The presentation of the data are confusing when the authors mix miR29b, HSP47 and TGF-B1 data together. It would be less confusing if the authors show step-wise effect of miR-29b, HSP47, and TGF-B1. The authors need to show the critical data that miR-29b directly targets HSP47, using reporter constructs and mutagenesis of miR-29b binding sites within HSP47. Also, having a schematic that illustrates the regulatory relationship between TGF-B1, miR-29b, HSP47, and EMT genes would be very helpful in guiding the reader. These are key points that need to be addressed prior to publication.

Other specific comments:
1. In the abstract, the sentence in lines 24 to 27 is not comprehensible. It would be good to be specifically stating the regulatory relationship between miR-29b of HSP47 and EMT genes. As written, it is unclear if miR-29 inhibits or promotes EMT genes.

  1. Introduction line 58: needs to provide references.
  2. Result section 2.1: as mentioned previously, the authors need to conduct experiment to demonstrate miR-29b’s direct suppression of HSP47 using reporter constructs and mutagenesis. Bioinformatic analysis and downstream changes of HSP47 and EMT genes do not prove that miR-29b directly regulates these genes and pathway.
  3. Results for Fig. 1B, 2A, 3A, 4A: It is critical that when using qPCR to measure the level of miR-29b that the authors use a relevant normalization control, such as another small RNA or U6. One cannot use GAPDH as an internal control. The internal control for measuring microRNA levels can be tricky, as it needs to be similarly expressed in control and experimental conditions and the length of the it needs to be more similar to the length of microRNAs.
  4. In Discussion, the sentence in line 202, what does ‘it’ refers to? It is not clear what the authors are referring to.
  5. Although the authors emphasized that the results of this study could potentially be used in clinical applications for treating CRS, what they showed would still be a big jump from here to the actual clinical application. Demonstrating miR-29 and HSP45 potentially can affect cell migration is still different from a whole animal work. Because of this, the authors should cut back emphasis on CRS, such as lines 209 to 227. Emphasis should placed on the basic understanding of how miR-29b regulates HSP in controlling EMT that is important in diseases such as CRS.
  6. The authors need to address what other targets miR-29b may target that potentially has other effects in addition to EMT. What other targets and biological processes miR-29b may regulate? This topic is important to discuss if miR-29b mimics were to be used in an animal.
  7. Another point is that one can design morpholinos to block miRNA binding site within a target gene. So once the authors define where miR-29b binds to HSP45, they can design this morpholino to block this regulation (Staton and Giraldez, Nature protocols, 2011). This could be a more specific way to block miR-29b’s suppression of HSP45.

Author Response

Reviewer: 1
Comments to the Author 

1) In the abstract, the sentence in lines 24 to 27 is not comprehensible. It would be good to be specifically stating the regulatory relationship between miR-29b of HSP47 and EMT genes. As written, it is unclear if miR-29 inhibits or promotes EMT genes.

Response: As you comment, I change the abstract.

“miR-29b mimic significantly inhibited the expression of HSP47 and TGF-β1-induced EMT-related markers in A549 cells. But, miR-29b inhibitor more induced the expression of them.”

2) Introduction line 58: needs to provide references.

Response: It is my mistake so, I change a reference.  

3) Result section 2.1: as mentioned previously, the authors need to conduct experiment to demonstrate miR-29b’s direct suppression of HSP47 using reporter constructs and mutagenesis. Bioinformatic analysis and downstream changes of HSP47 and EMT genes do not prove that miR-29b directly regulates these genes and pathway.

Response: As you comment, I showed that miR-29b directly targets HSP47 using luciferase assay. miR-29b mimic significantly inhibited the luciferase activity of the wild-type HSP47-3′-UTR. In contrast, miR-29b mimic had no effect on the luciferase activity of the mutant HSP47-3′-UTR. In addition to, miR-29b inhibitor significantly stimulated the luciferase activity of the wild-type HSP47-3′-UTR, however, miR-29b inhibitor had no effect on the luciferase activity of the mutant HSP47-3′-UTR. These date imply that miR-29b are directly interacted with the 3’-UTR of HSP47.

4) Results for Fig. 1B, 2A, 3A, 4A: It is critical that when using qPCR to measure the level of miR-29b that the authors use a relevant normalization control, such as another small RNA or U6. One cannot use GAPDH as an internal control. The internal control for measuring microRNA levels can be tricky, as it needs to be similarly expressed in control and experimental conditions and the length of that it needs to be more similar to the length of microRNAs.

Response: When I measured the level of miR-29b, I used both of GAPDH and U6 primer as normalization control, as I already mention in materials and methods section 4.4. However, I should mark U6 as normalization control in results as you mention. This is my mistake, so I fixed the figures.

5) In Discussion, the sentence in line 202, what does ‘it’ refers to? It is not clear what the authors are referring to

Response: I’m sorry but I could not find ‘it’ refers to’ in my manuscript.

6) Although the authors emphasized that the results of this study could potentially be used in clinical applications for treating CRS, what they showed would still be a big jump from here to the actual clinical application. Demonstrating miR-29 and HSP45 potentially can affect cell migration is still different from a whole animal work. Because of this, the authors should cut back emphasis on CRS, such as lines 209 to 227. Emphasis should placed on the basic understanding of how miR-29b regulates HSP in controlling EMT that is important in diseases such as CRS.

Response: We want to explain why we focus on tissue remodeling. We added contents that the miR-29b regulates HSP in controlling EMT that is important in diseases at 4th paragraph.

7) The authors need to address what other targets miR-29b may target that potentially has other effects in addition to EMT. What other targets and biological processes miR-29b may regulate? This topic is important to discuss if miR-29b mimics were to be used in an animal.

             Response: I added the text that explain the roles of miR-29b in discussion section. In addition to, it has been reported that miR-29b mimic has therapeutic effect of diseases with administration, injection or insert of miR-29b mimic in animal model. So we will consider it is necessary to confirm the effect of miR-29b mimics on tissue remodeling in mouse model.

8) Another point is that one can design morpholinos to block miRNA binding site within a target gene. So once the authors define where miR-29b binds to HSP45, they can design this morpholino to block this regulation (Staton and Giraldez, Nature protocols, 2011). This could be a more specific way to block miR-29b’s suppression of HSP45.

Response: Thank you for your comments. We will consider to use morpholinos to block miRNA binding site of target gene.

Reviewer 2 Report

I have reviewed the manuscript entitled” miR-29b Regulates TGF-β1-Induced Epithelial–Mesenchymal Transition by Inhibiting Heat Shock Protein 47 Expression in Airway Epithelial Cells” by Shin JM et al. Recent studies in chronic rhinosinusitis (CRS) suggest that the damage to the physical barrier in the sinonasal epithelium primes to a dysregulated immune response and pathologic tissue remodeling which might contribute to ongoing inflammation and disease refractoriness. In addition, epithelial-mesenchymal transition (EMT) plays an important role in dysregulated remodeling events and leads to refractory CRS, thus evidence has also shown that miR-29b regulates TGF-β1-induced EMT in a pulmonary fibrosis animal model. The authors have reported the relationship between the increased expression of HSP47 in nasal tissue and CRS severity in their previous study, they would like to profile the molecular regulatory association between miR-29b and HSP47 to TGF-b induced EMT which related with tissue remodeling in CRS in this paper.

They demonstrated that miR-29b modulates the protein and mRNA expression levels of EMT-related makers, which are induced by TGF-β1 in A549 cells, that process could be attenuated by HSP47 targeting siRNA. In addition, they also showed that TGF-β1-enhanced cell migration was significantly inhibited by the miR-29b mimic and siHSP47 in A549 cells, this study provides the first evidence that miR-29b suppresses TGF-β1-induced EMT and migration via HSP47 in airway epithelial cells. These results regarding the assessment of the connection to indicate that miR-29b and HSP47 are key regulators of TGF-β1-induced EMT in chronic airway inflammatory diseases such as CRS. Their data provides valuable information on the potential therapeutic targets of miR-29b and HSP47 in patients with refractory CRS is novel and interest. However, several minor comments can be made:

  1. Several cell-intrinsic signaling, such as TGF-β, WNT, and NOTCH pathways, are activated and cooperate to induce the expression of EMT during wound healing and tissue remodeling. Does TGF-β1 play the major role to induced the EMT in the airway epithelial cells? If the roles and the functions of HSP47 and miR-29b in EMT activity restricted to TGF-β1 signaling? Or, the regulatory networks between miR-29b and HSP47 is widespread action to EMT?
  2. Certain transcription factors, ZEB, SNAIL and TWIST that act pleiotropically to the cell state, are also showed the critical functions to response to EMT induction. If the expression of HSP47 and miR-29b could modulate these factors to react to TGF-β1 induced EMT?
  3. The expression profile of miR-29b might associate to the expression level of E-cadherin, and the appearance of other EMT related markers, a-SMA, Vimentin, and Fibronectin, were more closed to what HSP47 presented in this report. If any molecular mechanism acts on that association above?

Author Response

Reviewer: 2
Comments to the Author 

1) Several cell-intrinsic signaling, such as TGF-β, WNT, and NOTCH pathways, are activated and cooperate to induce the expression of EMT during wound healing and tissue remodeling. Does TGF-β1 play the major role to induce the EMT in the airway epithelial cells? If the roles and the functions of HSP47 and miR-29b in EMT activity restricted to TGF-β1 signaling? Or, the regulatory networks between miR-29b and HSP47 is widespread action to EMT?

Response: We previously reported that TGF-β1 has the potential to induce the EMT process in airway epithelial cells. We showed that the functions of HSP47 and miR-29b in EMT activity restricted to TGF-β1 signaling. However, it is possible that the regulatory networks between miR-29b and HSP47 is widespread action to EMT. In further study, we will investigate whether miR-29b and HSP47 regulate EMT process without TGF-β1.

2) Certain transcription factors, ZEB, SNAIL and TWIST that act pleiotropically to the cell state, are also showed the critical functions to response to EMT induction. If the expression of HSP47 and miR-29b could modulate these factors to react to TGF-β1 induced EMT?

Response: The previous study showed that miR-29b targeted snail in prostate cancer cells. So, miR-29b may target not only HSP47, but also EMT-related transcription factors such as snail. Although EMT-related transcription factors may be downstream signaling pathways of HSP47 and may regulate EMT as HSP47-independent pathway, we showed that HSP47 plays an important roles in regulating EMT.

3) The expression profile of miR-29b might associate to the expression level of E-cadherin, and the appearance of other EMT related markers, a-SMA, Vimentin, and Fibronectin, were more closed to what HSP47 presented in this report. If any molecular mechanism acts on that association above?

Response: As you mentioned, miR-29b and HSP47 may play different roles, but we think that both of miR-29b and HSP47 play important roles in regulating EMT because miR-29b directly targets HSP47. Since miR-29b can bind to various targets, it is necessary to find out which target gene is the most critical to target later.

Reviewer 3 Report

In the current manuscript Shin et al. investigated the role of miR-29b in the HSP47-regulated EMT of epithelial cells. The clinical challenge of the successful treatment of chronic rhinosinusitis gives a good and interesting basis for this work and the selected methods and techniques were carefully chosen to support the experiments. The manuscript is nicely written with only a few typos, however there are some misleading and hard-to-understand terms (like TGF-ß1-migrated) which should be changed.

On the other hand there are some major and serious issue in the Results section which have to be addressed before the manuscript could be accepted for publication.

1. In Results 2.1 the authors stated that "These results indicated that HSP47, a direct target of miR-29b, was induced by TGF-ß1 in A549 cells." This statement was based on an in silico prediction and experiments which investigated miR-29b and HSP47 mRNA levels seperately after TGF-ß1 treatment. At this point, there is no evidence of the direct regulation of HSP47 by miR-29b, because that could be a result of the treatment. I would suggest to remove or at least change this sentence, or provide more data to support this claim.

2. On Figure 2. the aim was to prove miR-29b overexpression could inhibit TGF-ß1 induced EMT, however the fold changes between miR control and miR control+TGF-ß1 is exactly the same as the change between miR-mimic control and miR-mimic+TGF-ß1 (1 to 0.5 vs 2 to 1, respectively). And the same is true for HSP47 levels. Furthermore, aSMA increases in control cells after miR-29b mimic transfection. Why?  And then after TGF-ß1 treatment why its getting lower compared to the same control? Vimentin worked very similarly. If the theory is the decrease in miR-29b results in the increase of HSP47 which then induce the EMT markers, why did the increase in miR-29b ended up in the increase of the EMT markers in untreated control cells? On the same figure the HSP47 immunostaining results are the opposite what the WB experiments show! Strong HSP47 signal could be observed in miR-29b overexpressing TGF-ß1 treated cells, while the WB showing a strong decrease compared to control. The changes in immunofluorescence maybe even stronger in miR-overexpressing cells than in controls. These results undermine the authors claim. Please, clarify or explain this phenomena!

3. On Figure 3. if miR-29b downregulating the HSP47 expression (based on the author's theory) than the inhibition of miR-29b why not increased HSP47 mRNA levels in untreated controls? miR control and miR-29b inhibitor cells should shown great difference in HSP47 expression, no? The question is the same for mRNA and protein expression results also! How the authors would solve this controversy?

4. On Figure 3. panel E the upper right image (+miR-29b-inhibitor+TGF-ß1 cells stained for HSP47/vimentin) is exactly the same as Figure 2. E upper right image (+miR-29b-mimic+TGF-ß1 cells stained for HSP47/vimentin)!!! Please correct it and include at least 3 separate representative images/cell+treatment in the supplement for all immunostaining experiments! I hope it's only a misunderstanding.

5. In Results 2.3 contains the following sentence: "In addition, TGF-ß1-downregulated E-cadherin upregulated by transfection of siHSP47 (Figure 4C and 4D)." There is literally no data to support this claim! mRNA levels (panel C) looking exactly the same before and after siHSP47 transfection and the protein measurements (panel D) showing the same! The above statement simply not true or at least not supported by any results which were cited!

Moreover, on the same figure, why did HSP47 levels decreased even further after TGF-ß1 treatment (panel D)? It should stay the same as the control, no? Additionally, how and why did HSP47 knockdown resulted in aSMA increase when in the non-transfected controls there aren't any aSMA (panel D)? Similarly, E-cadherin is still decreased after TGF-ß1 treatment which suggest HSP47 is not directly regulating EMT. Fibronectin also downregulated after TGF-ß1. Why?

On panel E no e-cadherin staining could be seen on the image, however the WB and mRNA data suggest that it should look like the siHSP47 sample! These issues have to be solved or explained very carefully!

Unfortunately, the presented results are confusing in many ways and this decreases the value of this work. After reading this manuscript I'm not convinced that miR-29b regulates HSP47 expression and EMT, rather, based on the presented data, I think it's more likely that TGF-ß1 treatment has a strong effect on the presented genes and markers. Please, address and explain the above listed issues.

Minor issues:

  1. Fig. 1 panel A is very low quality
  2. Fig1. D showing protein, not mRNA levels
  3. Fig2. C showing mRNA and not protein levels, also please change Conrol to Control
  4. Fig3. C showing mRNA and not protein levels
  5. Fig4. C showing mRNA and not protein levels
  6. In the Mat&Met section A549 cells were called a "human airways epithelial cell line" which is a misleading oversimplification, please include in the text that this is also a cancer cell line, therefore aberrant gene expression and altered molecular pathways could influence the results.
  7. Due to transfection is playing a key role in this work, please include some data (flow cytometry, image cytometry etc.) about the transfection efficiency for the experiments.

Author Response

Reviewer: 3
Comments to the Author 

1) In Results 2.1 the authors stated that "These results indicated that HSP47, a direct target of miR-29b, was induced by TGF-ß1 in A549 cells." This statement was based on an in silico prediction and experiments which investigated miR-29b and HSP47 mRNA levels separately after TGF-ß1 treatment. At this point, there is no evidence of the direct regulation of HSP47 by miR-29b, because that could be a result of the treatment. I would suggest to remove or at least change this sentence, or provide more data to support this claim.

Response: As you comment, I showed that miR-29b directly targets HSP47 using luciferase assay. miR-29b mimic significantly inhibited the luciferase activity of the wild-type HSP47-3′-UTR. In contrast, miR-29b mimic had no effect on the luciferase activity of the mutant HSP47-3′-UTR. In addition to, miR-29b inhibitor significantly stimulated the luciferase activity of the wild-type HSP47-3′-UTR, however, miR-29b inhibitor had no effect on the luciferase activity of the mutant HSP47-3′-UTR. These date imply that miR-29b are directly interacted with the 3’-UTR of HSP47.

2) On Figure 2. the aim was to prove miR-29b overexpression could inhibit TGF-ß1 induced EMT, however the fold changes between miR control and miR control+TGF-ß1 is exactly the same as the change between miR-mimic control and miR-mimic+TGF-ß1 (1 to 0.5 vs 2 to 1, respectively). And the same is true for HSP47 levels. Furthermore, aSMA increases in control cells after miR-29b mimic transfection. Why?  And then after TGF-ß1 treatment why its getting lower compared to the same control? Vimentin worked very similarly. If the theory is the decrease in miR-29b results in the increase of HSP47 which then induce the EMT markers, why did the increase in miR-29b ended up in the increase of the EMT markers in untreated control cells? On the same figure the HSP47 immunostaining results are the opposite what the WB experiments show! Strong HSP47 signal could be observed in miR-29b overexpressing TGF-ß1 treated cells, while the WB showing a strong decrease compared to control. The changes in immunofluorescence maybe even stronger in miR-overexpressing cells than in controls. These results undermine the authors claim. Please, clarify or explain this phenomena!.

Response: In raw data, the fold changes between miR control and miR control+TGF-ß1 is not exactly the same as the change between miR-mimic control and miR-mimic+TGF-ß1 (1.04 to 0.51 vs 2.23 to 0.83, respectively). And, in HSP47 data, the fold changes between miR control and miR control+TGF-ß1 is not exactly the same (1.02 to 1.98 vs 0.65 to 0.98, respectively). In addition to, as you mentioned, we also expected that c and vimentin expression were decreased in control with miR-29b mimic. Our data showed that that α-SMA and vimentin expression were increased in control with miR-29b mimic compared with control with miR-29b control but it not statistically significant. In immunostaining results, I put in the wrong data. So I fixed it.

3) On Figure 3. if miR-29b downregulating the HSP47 expression (based on the author's theory) than the inhibition of miR-29b why not increased HSP47 mRNA levels in untreated controls? miR control and miR-29b inhibitor cells should shown great difference in HSP47 expression, no? The question is the same for mRNA and protein expression results also! How the authors would solve this controversy?

Response: As you mentioned, we also expected that miR-29b inhibitor increased HSP47 mRNA levels in untreated controls. But, we showed that miR-29b inhibitor did not affect HSP47 mRNA levels in untreated controls. So, we measured HSP47 mRNA levels in controls + miR-29b inhibitor. It seem that miR-29b inhibitor increased HSP47 mRNA levels but it is not statistically significant. It may imply that miR-29b more affect TGF-ß1-indcued HSP47.

4) On Figure 3. panel E the upper right image (+miR-29b-inhibitor+TGF-ß1 cells stained for HSP47/vimentin) is exactly the same as Figure 2. E upper right image (+miR-29b-mimic+TGF-ß1 cells stained for HSP47/vimentin)!!! Please correct it and include at least 3 separate representative images/cell+treatment in the supplement for all immunostaining experiments! I hope it's only a misunderstanding

Response: I am sorry. It is my mistake. As I answered the above question, I put the wrong data in the process of editing the data to put it in the same position. So I change the data.

5) In Results 2.3 contains the following sentence: "In addition, TGF-ß1-downregulated E-cadherin upregulated by transfection of siHSP47 (Figure 4C and 4D)." There is literally no data to support this claim! mRNA levels (panel C) looking exactly the same before and after siHSP47 transfection and the protein measurements (panel D) showing the same! The above statement simply not true or at least not supported by any results which were cited!

Response: I checked whether TGF-ß1-downregulated E-cadherin mRNA upregulated by transfection of siHSP47 again. mRNA and ICC data showed that TGF-ß1-downregulated E-cadherin upregulated by transfection of siHSP47.

6) Moreover, on the same figure, why did HSP47 levels decreased even further after TGF-ß1 treatment (panel D)? It should stay the same as the control, no? Additionally, how and why did HSP47 knockdown resulted in aSMA increase when in the non-transfected controls there aren't any aSMA (panel D)? Similarly, E-cadherin is still decreased after TGF-ß1 treatment which suggest HSP47 is not directly regulating EMT. Fibronectin also downregulated after TGF-ß1. Why?

Response: It seems that α-SMA and HSP47 levels decreased in TGF-ß1 + siHSP47 compared with control. But it is not statistically significant. Like the answer above, mRNA and ICC data showed that TGF-ß1-increased α-SMA and HSP47 levels did not decreased by transfection of siHSP47 compared with control.

7) Fig. 1 panel A is very low quality

Response: Thank you, I fixed Fig.1 panel A.

8) Fig1. D showing protein, not mRNA levels

Response: I fixed Fig 1D, mRNA to protein

9) Fig2. C showing mRNA and not protein levels, also please change Conrol to Control

Response: I fixed Fig 2D, mRNA to protein and Conrol to Control.

10) Fig3. C showing mRNA and not protein levels

Response: I fixed Fig 3D, mRNA to protein and Conrol to Control.

11) Fig4. C showing mRNA and not protein levels

Response: I fixed Fig 4C, mRNA to protein and Conrol to Control.

12) In the Mat&Met section A549 cells were called a "human airways epithelial cell line" which is a misleading oversimplification, please include in the text that this is also a cancer cell line, therefore aberrant gene expression and altered molecular pathways could influence the results.

Response: As you mentioned I change the manuscript in mat&met.

“The human airway epithelial cell line A549 (ATCC CCL-185) was obtained from the American Type Culture Collection, Rockville, MD, USA. A549 is a cancer cell line, therefore aberrant gene expression and altered molecular pathways could influence the results”

13) Due to transfection is playing a key role in this work, please include some data (flow cytometry, image cytometry etc.) about the transfection efficiency for the experiments..

Response: Thank you for your comment. If other data is needed, it will be added later. However, I think that the transfection efficiency has been sufficiently verified by checking the mRNA level.

Round 2

Reviewer 3 Report

While I'm thankful for the authors for the luciferase assay experiments, most of my serious questions left unanswered or directly avoided. This study could be an important one, however the results are often misinterpreted and controversial with no provided solution for the issues.

Some examples:

  • No matter whether miR-29b mimic or miR-29b inhibitor was transfected into the cells, aSMA and vimentin (and maybe even fibronectin too) levels increased compared to the parental cell line. Because these constructs should have the opposite effects at any time, the results suggest that's the effect of transfection itself rather than the constructs.
  • As I wrote in the first round the sentence "In addition, TGF-ß1-downregulated E-cadherin upregulated by transfection of siHSP47 (Figure 4C and 4D)." is baseless because on this figure no results supporting this claim. The authors asnwered "mRNA and ICC data showed that TGF-ß1-downregulated E-cadherin upregulated by transfection of siHSP47." mRNA clearly shows no difference and maybe there is something in the ICC results, however due to the western blot data the protein results are also questionable. At this point I will leave this to the Editor to decide because none of my colleagues neither me were able to find the data which, at least, would slightly suggest that this statement is true.
  • The authors corrected several minor issues (like mRNA rather then protein and Conrol to Control), however even in the latest version of the manuscript these small mistakes were there. These have to be corrected in the final version!
  • I asked for data showing the transfection efficiency, however the only answer I got was "Thank you for your comment. If other data is needed, it will be added later. However, I think that the transfection efficiency has been sufficiently verified by checking the mRNA level." This response is fundamentally wrong. Checking the mRNA levels gives only data about the expression on the populaton level which could come from two seperate sources. 1. All the cells in the population were transfected equally and therefore express the genes similarly. This rarely happens. 2. Only a fraction, a subpopulation of the cells were transfacted and these cells express high levels of the genes. It's an important question to answer because uneven transfection of the cells could explain the anomaly seen throughout the manuscript. To be honest, it's quite strange that a reviewer has to ask for this data because in better journals these results have to be included at the moment when the manuscript submitted.

Overall the work is good, the experiments are nicely done, but the drawn conclusions and the machanistic link between miR-29b, HSP47 and EMT is not fully supported. I don't want to reject this work, so I will suggest another round of revision, but these issues have to be solved before the manuscript could be accepted for publication.

Author Response

Reviewer: 3
Comments to the Author 

1) No matter whether miR-29b mimic or miR-29b inhibitor was transfected into the cells, aSMA and vimentin (and maybe even fibronectin too) levels increased compared to the parental cell line. Because these constructs should have the opposite effects at any time, the results suggest that's the effect of transfection itself rather than the constructs.

Response: Thanks for your comment. I also re-checked the effect of miR-29b mimic on E-cadherin, fibronectin, α-SMA, and HSP47 mRNA and protein levels. When I transfected miR-29b mimic into the cells, α-SMA, vimentin and fibronectin levels were decreased in miR-29b mimic + Control compared with miR Control + Control. These data showed that HSP47 was decreased by the added miR-29b and EMT was suppressed, but it was not statistically significant. However, when miR-29b inhibitor was transfected into the cells, as shown in the previous data, expression of α-SMA and vimentin increased statistically insignificantly. It is thought to be because miR-29b present in control cells was inhibited.

2) As I wrote in the first round the sentence "In addition, TGF-ß1-downregulated E-cadherin upregulated by transfection of siHSP47 (Figure 4C and 4D)." is baseless because on this figure no results supporting this claim. The authors asnwered "mRNA and ICC data showed that TGF-ß1-downregulated E-cadherin upregulated by transfection of siHSP47." mRNA clearly shows no difference and maybe there is something in the ICC results, however due to the western blot data the protein results are also questionable. At this point I will leave this to the Editor to decide because none of my colleagues neither me were able to find the data which, at least, would slightly suggest that this statement is true.

Response: Thanks for your comment. I also re-checked the effect of siHSP47 on E-cadherin, fibronectin, α-SMA, and HSP47 mRNA and protein levels. In this experiment data, E-cadherin upregulated by transfection of siHSP47 in TGF-ß1-stimulated A549 cells.

3) I asked for data showing the transfection efficiency, however the only answer I got was "Thank you for your comment. If other data is needed, it will be added later. However, I think that the transfection efficiency has been sufficiently verified by checking the mRNA level." This response is fundamentally wrong. Checking the mRNA levels gives only data about the expression on the populaton level which could come from two seperate sources. 1. All the cells in the population were transfected equally and therefore express the genes similarly. This rarely happens. 2. Only a fraction, a subpopulation of the cells were transfacted and these cells express high levels of the genes. It's an important question to answer because uneven transfection of the cells could explain the anomaly seen throughout the manuscript. To be honest, it's quite strange that a reviewer has to ask for this data because in better journals these results have to be included at the moment when the manuscript submitted.

Response: As you said, I tried to measure the transfection efficiency using flow cytometry. When siRNA-GFP was transfected, the efficiency was about 95%. When HSP47 was measured after transfection with siHSP47, about 62% was inhibited.
